# A Novel Meander Line Metamaterial Absorber Operating at 24 GHz and 28 GHz for the 5G Applications

**DOI:** 10.3390/s22103764

**Published:** 2022-05-15

**Authors:** Syed Aftab Naqvi, Muhammad Abuzar Baqir, Grant Gourley, Adnan Iftikhar, Muhammad Saeed Khan, Dimitris E. Anagnostou

**Affiliations:** 1Department of Electrical and Computer Engineering, COMSATS University Islamabad, Sahiwal 57000, Pakistan; aftabnaqvi@cuisahiwal.edu.pk (S.A.N.); adnaniftikhar@comsats.edu.pk (A.I.); 2Institute of Signals, Sensors and Systems, Heriot Watt University, Edinburgh EH14 4AS, UK; gjg1@hw.ac.uk; 3Department of Information Engineering, University of Padova, 35122 Padova, Italy; mskj786@hotmail.com

**Keywords:** meander line, metamaterial absorber, massive MIMO, 5G

## Abstract

Fifth generation (5G) communication systems deploy a massive MIMO technique to enhance gain and spatial multiplexing in arrays of 16 to 128 antennas. In these arrays, it is critical to isolate the adjacent antennas to prevent unwanted interaction between them. Fifth generation absorbers, in this regard, are the recent interest of many researchers nowadays. The authors present a dual-band novel metamaterial-based 5G absorber. The absorber operates at 24 GHz and 28 GHz and is composed of symmetric meander lines connected through a transmission line. An analytical model used to calculate the total number of required meander lines to design the absorber is delineated. The analytical model is based on the total inductance offered by the meander line structure in an impedance-matched electronic circuit. The proposed absorber works on the principal of resonance and absorbs two 5G bands (24 GHz and 28 GHz). A complete angular stability analysis was carried out prior to experiments for both transverse electric (TE) and transverse magnetic (TM) polarizations. Further, the resonance conditions are altered by changing the substrate thickness and incidence angle of the incident fields to demonstrate the functionality of the absorber. The comparison between simulated and measured results shows that such an absorber would be a strong candidate for the absorption in millimetre-wave array antennas, where elements are placed in proximity within compact 5G devices.

## 1. Introduction

The exponential proliferation of frequency-selective devices (including complex wireless electronic systems and internet of things (IoT), etc.) demand large bandwidths, high data rates, and low latencies to operate efficiently and purposefully. The 5G telecommunication network system is the proposed solution to this problem, providing a bandwidth of around 5 GHz, a data rate of up to 5 Gb/s for high mobility and 50 Gb/s for pedestrians, and a low latency of 1 ms [1,2]. The availability of such a wide bandwidth in the mm-wave band was approved in August 2018 by the Federal Communications Commission (FCC) for 5G, during the first 5G spectrum auction for the 24 GHz (24.25–24.45 and 24.75–25.25) and 28 GHz (27.5–28.35 GHz) bands. This allowed researchers to focus on new designs at these frequencies [3].

Novel metamaterial absorbers (MAs), which work on the principal of resonance, have been explored for use with wireless communication devices (emitters, filters, sensors, photodetectors, photovoltaic solar cells, and infrared camouflage) [4,5,6,7,8,9,10,11,12]. Significant work on MA designs is available in the literature for microwave, terahertz, visible, and ultraviolet frequencies [13,14,15,16,17,18]. On the contrary, 5G and especially 24 GHz and 28 GHz bands are relatively unattended so far, in this regard [19,20,21,22]. Additionally, the limitations of previous works on narrowband, wideband, and ultra-wideband absorbers involve the facts of their complex geometrical structures, large number of layers to trap electromagnetic waves, and costly materials [13,14,15,16,17,18,23,24,25,26].

A meander line is a U-shaped compact transmission line (TL), formed by connecting two parallel TLs with another TL [27,28]. Meander lines achieve compactness as they offer similar impedance Z, inductance L, and other microwave network parameters to straight planar TLs [27,28,29,30]. Their reduced overall size also allows them to utilize the given design space more effectively. The total electrical length of the meander line is similar to that of a straight line; hence, the operating frequency remains the same. The ease of integration of a meander line with the rest of the electronic circuitry, as well as the possibility of shifting its frequency by varying the number of meanders, are extremely useful features of meander lines [31]. Meander lines are frequently used in antennas; however, there are very few examples of absorbers employing meander lines.

In this paper, a novel dual-band absorber operating at 24 GHz and 28 GHz for 5G applications is proposed. The material used to design the absorber is cost-effective and the structure is easy to design when compared to the absorbers presented in the literature to date. Novelty lies in the fact that this is the very first attempt to design an absorber for 5G applications at the 24 GHz and 28 GHz frequency bands. Moreover, the derivation of the number of meander lines to devise a structure for the absorber at any frequency by varying the parameters of the meander line is a unique feature of the proposed work in this domain. The utilization of the meander line structures to design an absorber at millimetre-wave frequency is an attempt which is the first of its kind, to the best of the authors’ knowledge.

The analytical derivation used to calculate the number of meander lines in order for the absorber to operate at 24 GHz and 28 GHz, as well as the overall design procedure, is presented in Section 2. The simulated and measured results of the absorber are discussed in Section 3. Finally, the conclusion is discussed in Section 4.

## 2. Design Procedure and Absorption Mechanism

Figure 1b shows the front view of the unit cell of the proposed metamaterial absorber. The absorber consists of two pairs of symmetric meander lines connected to an I-shape TL. The schematic representation of the finite absorber sheet consisting of the proposed metamaterial absorber is depicted in Figure 1a. The procedure for calculating the total number of meander lines needed to design the proposed novel metamaterial absorber is delineated here. This will help the reader and scientific community in designing on-demand metamaterial absorbers for their respective applications in numerous disciplines, according to the requirements of the end user. The procedure for calculating the number of meander lines at the specific frequencies is shown below:

The characteristic impedance of the parallel placed twin TL is [32]
(1)$$Z_{0}=\frac{\eta }{\pi }\log\frac{2a}{b}\;$$
where *η* is the intrinsic impedance of the free space, *a* is the distance between the lines, and *b* is the diameter of a TL. If *l* is the length of a line terminated with a load impedance *Z_L_*, then the input impedance of the twin TL is given by [28]
(2)$$Z_{in}=Z_{0}\frac{Z_{L}+jZ_{0}\tan\beta l}{Z_{0}+jZ_{L}\tan\beta l}$$
where $\beta =\frac{2\pi }{\lambda }$, $\lambda =\frac{c}{f\sqrt{\epsilon_{r}}}$, c is the speed of light, f the operating frequency, and *ε_r_* the relative permittivity of the material. *Z_in_* is the input impedance of the short-terminated lines which will help to propagate the matched impinged EM waves on the structure. Meander lines are truncated with a short-circuit load (*Z_L_* = 0), so (2) becomes
(3)$$Z_{in}=jZ_{0}\tan\beta l\;$$

The resultant *Z_in_* is a pure reactance. Short-terminated meander lines are inductive loads [29], so *Z_in_= j**ω**L*, where *ω* is the angular frequency and *L* is the equivalent inductance. Moreover, at high frequencies where βl<1, series expansion up to the third order is used, and (3) is further simplified as
(4)$$Z_{in}=j\omega L=jZ_{0}\left(\beta l+\frac{1}{3}\;{\left(\beta l\right)}^{3}\right)\;$$

The formation of the current on the surface of the structure will be at its maximum if all the components of the structure have the same impedance. So, all the meanders connected to the transmission line must have the same impedance to maximally absorb and distribute the imping EM waves on the surface of the structure with minimum reflection. For *N* numbers of meander line inductors, the total impedance is *NZ_in_ = j**ω**L_m_*. Here, *L_m_* is the total inductance of all the short-terminated lines. Now, substituting the values in (4) gives
(5)$$j\omega L_{m}=NZ_{in}=jNZ_{0}\left(\beta l+\frac{1}{3}\;{\left(\beta l\right)}^{3}\right)$$

Then, substituting *Z*_0_ from (1), $\beta =\omega \sqrt{\mu \epsilon }$ and $\eta =\sqrt{\frac{\mu }{\epsilon }}$, *L_m_* can be written as: (6)$$L_{m}=\frac{N\mu l}{\pi }\log\frac{2a}{b}\left(1+\frac{1}{3}\;{\left(\beta l\right)}^{2}\right)\;$$

Next, if $l^{\prime }\;$ is the total length of the meander lines, then self-inducting *L_l_* is mathematically defined as [33]:(7)$$L_{l}=\frac{\mu }{\pi }l^{\prime }\left(\log\left(\frac{4l^{\prime }}{b}\right)-1\right)\;$$

So, the total inductance of a meander line can be expressed by combining (6) and (7):(8)$$L_{t}=\frac{N\mu l}{\pi }\log\frac{2a}{b}\left(1+\frac{1}{3}\;{\left(\beta l\right)}^{2}\right)+\frac{\mu }{\pi }l^{\prime }\left(\log\left(\frac{4l^{\prime }}{b}\right)-1\right)\;\;$$

Now, if a dipole and a meander line are operating at the same frequency, then the inductances of the two should also be same. Let *L_d_* be the self-inductance of a wire, so that is defined as [32]
(9)$$L_{d}=\frac{\mu }{\pi }\frac{\lambda }{4}\left(\log\left(\frac{l^{\prime }}{b}\right)-1\right)\;$$

Finally, comparing (8) and (9), the number of turns *N* can be calculated mathematically:(10)$$N=\frac{\frac{\lambda }{4}\left(\log\left(\frac{l^{\prime }}{b}\right)-1\right)-l^{\prime }\left(\log\left(\frac{4l^{\prime }}{b}\right)-1\right)}{l\;\log\frac{2a}{b}\left(1+\frac{1}{3}\;{\left(\beta l\right)}^{2}\right)}$$
if $l\leq \frac{\lambda }{2}$, then (10) can be further simplified by considering only the first progression of the series expansion, which results in a simpler form:(11)$$N=\frac{\frac{\lambda }{4}\left(\log\left(\frac{l^{\prime }}{b}\right)-1\right)-l^{\prime }\left(\log\left(\frac{4l^{\prime }}{b}\right)-1\right)}{l\;\log\frac{2a}{b}\;}\;$$

Equation (11) gives the total number of turns required to design an absorber at 24 GHz and 28 GHz. The equation also deliberates the relationship of the spacing between meander lines and frequency, i.e., an increase in meander separation will decrease the resonant frequency and vice versa by keeping the rest of the parameters the same. So, one can obtain the required meanders and thus novel structure by tuning the parameters given in (11).

By inserting the values of the parameters as shown in Figure 1b, the obtained number of turns are 3.30 and 3.35 at 24 GHz and 28 GHz, respectively. Consequently, the closest larger integer of the four meander lines is chosen in consideration of the absorber geometry to achieve unity absorption at the band of interest. Based upon the above analytically extracted geometrical parameters, the absorber shown in Figure 1 is designed and then simulated in CST Microwave Studio. The design consists of two symmetric pairs of meander lines (four meander lines) connected with an I-shape TL, resulting in the simple resonating structure depicted in Figure 1a. The design is realized on a 1.6 mm thick FR-4 substrate, with *ɛ**_r_* = 4.4 and loss tangent tan*δ* = 0.02. In CST, the unit cell boundary conditions are deployed along the *x*- and *y*-axis, whereas the open boundary conditions are deployed along the *z*-axis. Further, both the TE- and TM-polarized waves propagating along the *z*-axis are obliquely excited on the metamaterial structure to observe the absorption behaviour of the proposed absorber at the frequency band.

### Absorption Mechanism

The absorber is composed of the top resonating surface on the dielectric substrate, backed with a copper metallic layer. The top resonating surface allows for the penetration of the incoming microwave when the resonating conditions have been satisfied. The bottom metallic layer blocks this transmission of the microwave. The middle FR4 lossy dielectric layer binds to the electromagnetic wave and converts it into another form of energy. The resonance on the top metasurface is attained by satisfying the free-space impedance, matching with the absorber impedance. As the absorber impedance matches the free-space impedance, all incidence light gets absorbed and reflection is zero, i.e., ${\left|S_{11}\right|}^{2}\approx 0$.

## 3. Results and Discussions

The absorptivity behaviour of the proposed microwave absorber (MA) was observed in the simulation software using the parametric analysis of the geometrical parameters. Initially, the absorptivity behaviour was analysed by changing the thickness of the dielectric substrate at *t* = 0.8 mm, 1.2 mm, and 1.6 mm. Then, the effects of the obliquity incidence wave on the absorptivity of the proposed MA were observed for both TE and TM polarizations. To better understand the absorption mechanism of the proposed MA, the normalized impedance and surface electric field were also studied to understand the peak absorption of the proposed MA.

Figure 2 illustrates the absorptivity versus the frequency of the MA with a substrate thickness of *t* = 0.8 mm from 0° to 30° of both TE- and TM-polarized incident waves. Figure 2a shows the absorption for TE polarization corresponding to different angles of incidence from θ= 0° to 30° (with step size of 10°). It is depicted that for θ = 0°, an absorption peak is attained at 26 GHz (as shown in Figure 2a). The results show that with the increase in incidence angle, the absorption peak produces a redshift. Further, it is observed that absorption peaks are attained at 26 GHz, 25.8 GHz, 25.6 GHz, and 25.3 GHz for their respective angles of incidence at θ = 0°, 10°, 20°, and 30°. Notably, for θ = 30°, the proposed MA is most useful in absorbing the 5G band ranging from 24.57 GHz to 25.25 GHz, having a full width half maximum (FWHM) of 0.8 GHz. This shows that the MA covers the entire 5G band. Figure 2b shows the absorption for TM polarization, keeping angular operating conditions like the previously discussed case in Figure 2a. Moreover, it is noticeable that absorption behaviour is changed by altering the angle of incidence. However, for θ > 20°, similar absorptivity is attained as observed for the TE-polarized case depicted in Figure 2a. From this, it can be inferred that for θ > 20°, MA acts as a 5G absorber.

Figure 3 illustrates absorption when the substrate thickness is increased to *t* = 1.2 mm. It is noticeable that with the increase in substrate thickness, absorption spectra are altered, compared with the previous case discussed in Figure 2. Figure 3a illustrates the absorptivity of the MA for a TE-polarized wave. Absorption peaks show a redshift with the increase in incidence angle for TE-polarized waves. It is observed that for incidence angles of θ = 20° and 30°, the MA covers the 5G bands (27.5 GHz–28.35 GHz), whereas, for excitation θ = 10°, another 5G band (24.75 GHz–25.25 GHz) is absorbed by the proposed MA. In addition, it is noticed that the highest absorption peak with near-unity absorption is attained for θ = 20°, just before 24 GHz, shown by dash-dotted blue line. In examining the second absorption peaks at higher frequencies, it is observed that these absorption peaks cover another 5G band (27.5 GHz–28.35 GHz) for every incident angle from 0° to 30° (as shown in Figure 3a). The absorption has a large value of FWHM in this case. Moreover, absorptivity increases with the increase in incidence angle. It is noticed that for θ = 0°, the absorption peak attains absorptivity of 75%, and for θ = 30°, the absorption peak has absorptivity of 95%. Additionally, considering Figure 3b, it can be concluded that for θ = 0°, 10°, and 20° in TM-polarized cases, absorptivity lies within the 5G band (24.25 GHz–24.45 GHz and 24.75 GHz–25.24 GHz) with more than 92% absorption.

The absorption peaks of the proposed MA are studied for thickness of the substrate at *t* = 1.6 mm. Figure 4 represents the simulated absorptivity of the chosen MA design. It is observed that with the increase in substrate thickness, absorptivity shows a considerable increase. Further, it can be clearly seen that the number of absorption peaks are increased, as compared to the previous cases discussed in Figure 2 and Figure 3. Figure 4a shows the absorptivity for the TE-polarized wave. Here, it is observed that for θ = 0°, 10°, 20°, and 30°, MA successfully covers the 24 GHz (i.e., 24.25 GHz–24.45 and 24.75 GHz–25.25 GHz) band, with absorption higher than 90% for all values of θ. On the other hand, for the 28 GHz 5G band, it is observed that absorption peaks lie in this band for θ = 0° and 10°, as shown by black solid and dashed red lines in Figure 4a. However, the absorption peak shifts to the higher frequency for θ = 20°, as shown by the dashed-dotted blue line in Figure 4a, whereas, for θ = 30°, the absorption peak is attained with an absorptivity of 62% just before 28 GHz (as shown by the dashed green line in Figure 4a). Figure 4b shows the absorptivity for the TM polarization. It is observed that the absorption peaks are found in the 28 GHz frequency range of the 5G—the remaining absorption peaks are not in our interest.

To create a better understanding of the absorption mechanism, the surface current density of the top metasurface and bottom layer is depicted in Figure 5 and Figure 6, corresponding to the absorption peaks at 24.7 and 28.3 GHz, respectively. Figure 5 shows the surface current density for the absorption at 24.7 GHz, and it is noticeable that the maximum surface current remains confined between the top and lower sides of the adjacent mender lines and the middle of the transmission line. Furthermore, the direction of the surface currents is parallel for the top metasurface and bottom conductor in most parts of the unit cell, which leads to electric resonance. However, the surface current on the mender line is antiparallel to the surface current of the bottom layer, which leads to the magnetic dipole. Hence, the absorption is due to the electric and magnetic resonance. Similarly, the surface current density plots at 28.3 GHz are shown in Figure 6, which reveals the formation of electric and magnetic dipoles at the surface of the meander line and the resultant phenomenon of absorption. 

The impedance was also studied to further explore the physical mechanism of the absorption phenomena. Figure 7 illustrates both the real and imaginary effective impedance for θ = 0°, keeping the substrate thickness as *t* = 1.6 mm. Herein, the impedance-matching condition is considered. It is noteworthy that due to the resonance, the free-space impedance matches the top metasurface impedance; therefore, all incidence waves are allowed to enter in the substrate of the metamaterial absorber. The effective impedance of the proposed MA is deduced by employing the impedance formula [18]. The plot depicted in Figure 7 shows the effective impedance for a normal-incidence TE-polarized wave with a dielectric substrate thickness of 1.6 mm. It is noticeable that the real part of the effective impedance shows a sharp dip at ~24.5 GHz and ~28.2 GHz (in the 5G band allocated by FCC), confirming that the proposed MA absorbs frequencies within this band. The imaginary effective impedance attains negative values at the aforesaid frequencies: impedance at ~24.5 GHz and ~28.2 GHz matches with the free-space impedance, allowing the electromagnetic waves to penetrate within the dielectric substrate. Thereby, all the incidence waves remain trapped inside the dielectric substrate around the reported frequencies. Consequently and corresponding to the impedance-matching condition, two absorption peaks are observed in Figure 4a for the normally incident excited wave.

To further investigate the absorption of the proposed metamaterial absorber, the surface electric field is studied with the absorption peak at 24.5 GHz for a normally incident wave. Figure 8a,b illustrate the surface electric field for the TE and TM polarizations, respectively. It is noticeable from the figures that the electric field is maximally concentrated around the edges of the meander line. It can also be observed that the transmission line of the meander line absorber has a lower concentration of electric field, as presented in Figure 8.

To verify the simulation results, an array of 12 × 12 unit cells was fabricated on FR4 dielectric substrate with copper on its bottom side. Although the simulation results of *t* = 0.8 mm and *t* = 1.2 mm are encouraging, a substrate thickness of 1.6 mm was chosen for the fabrication of the proposed absorber because of its commercial availability. A photograph of the measurement setup and the finite MA consisting of 12 × 12 unit cells of the proposed meander-line-based absorber printed on 1.6 mm thick FR4 substrate is shown in Figure 9. In the measurement setup, initially, the system was calibrated by placing a metallic sheet in front of the transmitter and receiver horn antennas which were connected to a well-calibrated vector network analyser (VNA). It was also ensured that the horn antennas were placed at a far-field distance from the MA to avoid electromagnetic inference which may have altered the results. 

After the system was calibrated, the fabricated MA was placed in front of the horn antennas, and the magnitude of the transfer coefficient (|S_21_| (dB) was measured using the VNA for a TE-polarized wave. Then, the MA sheet was rotated horizontally (along the azimuthal plane) from 0° to 30° with a step size of 10°, and the (|S_21_| (dB) was recorded on the VNA. Similarly, the (|S_21_| (dB) was recorded by rotating the horn antennas by 90° for a TM polarization and by rotating the sheet from 0° to 30° in azimuth. The recorded values of the (|S_21_| (dB) were then post processed to calculate the absorption peaks over the frequency range using 1 – |S_11_|^2^ – |S_21_|^2^ [32]. The |S_21_| is the wave received by the receiver horn antenna, after impinging on the proposed MA from the transmitting horn. The measured |S_11_|≈ 0, here |S_11_| is the reflected EM wave by the respective horn antenna, on a linear scale. Figure 10 shows the comparison of the simulated and measured results for the TE wave. For the sake of clarity, the comparison of results for different incidence angles is presented separately.

Figure 10a presents the results of when the incident angle is 0°. From this, it is evident that the absorber acts as a dual-band one, operating in the 5G regions of 24 GHz and 28 GHz, i.e., one peak lies in 24.25 GHz–24.45 GHz and the other in the 27.5 GHz–28.35 GHz region. Moreover, the absorptivity is more than 92% over these same frequency bands in both simulated and measured results. Figure 10b shows the comparison for θ = 10°. Again, there are two peaks in the 5G frequency bands, namely 24 GHz (24.75 GHz–25.25 GHz) and 28 GHz (27.5 GHz–28.35 GHz). The simulated and measured results are both in good agreement with each other and, overall, absorptivity is greater than 90% for both.

Figure 10c depicts that at θ = 20°, the proposed absorber absorbs a single 5G frequency band. It is noticeable that at this specific incidence angle for a TE polarization, the proposed absorber exhibits around 95% absorptivity in the 24.75 GHz–25.25 GHz frequency region of 5G. Figure 10d reflects the absorption at θ = 30°, and dual-band absorption behaviour in 5G frequency bands can also be noticed here. One absorption peak is observed in the allocated 5G band ranging from 24.75 GHz to 25.25 GHz with near-unity absorptivity in its measurements, while the other peak is observed in the 28 GHz (27.5 GHz–28.35 GHz) frequency band with around 75% measured absorptivity. Overall, both the simulated and measured absorption results of the proposed absorber in the 5G frequency bands are in good agreement at different obliquity angles for TE polarization. 

Figure 11 shows the comparison of simulated and measured results when the TM mode is excited on the proposed absorber. Again, the comparison of results for the different incidence angles is presented separately for clarity. It is noticeable that unlike for the TE-polarized wave case, the proposed absorber here behaves like a single-band absorber when polarization is TM. To explain, it either behaves like an absorber in the 24 GHz or one in the 28 GHz frequency band for different angles of incidence.

Figure 11a shows the behavior of the absorber at 24 GHz (24.25 GHz–24.45 GHz) with a 98% absorptivity when the incident angle is 0°, whilst more than 95% absorptivity is observed at the 28 GHz (27.5 GHz–28.35 GHz) frequency band for θ = 10°, as depicted in Figure 11b. Both the simulated and measured results for incidence angles of θ = 20° and 30° are shown in Figure 11c,d, respectively, where, again, the results at 28 GHz (27.5 GHz–28.35 GHz) are reported as an absorptivity of around 85% and 80%. It is reported, again, that the overall simulated and measured results are in good agreement (within the respective 5G band) with each other. Ripples at different frequency bands other than the 5G ones are attributed to the imperfect fabrication of the absorber sheet and the use of RF cables within the anechoic chamber.

## 4. Conclusions

A metamaterial-based absorber operating at 24 GHz and 28 GHz frequency bands and allocated for 5G applications by the FCC was investigated here. The metamaterial structure consists of four meander lines connected with an I-shaped TL. The number of meander lines necessary to design the absorber at the proposed frequency bands was calculated using an analytical formulation, based on the phenomenon of total inductance produced by a meander line structure, under the matched-impedance conditions in the given electronic circuit at the operating frequencies. The meander line structure, created using this analytical formulae, was used to design the resultant structure and was then optimized and simulated in CST to demonstrate the efficiency of the analytical model along with the performance of the absorber. A complete parametric analysis was carried out to demonstrate the design flexibility of the proposed absorber. To explore the practical employability and verify the simulated results of the absorber, a finite sheet of the proposed MA, with 12 × 12 unit cells, was fabricated and tested in an anechoic chamber. In addition, the simulation results of the proposed MA for TE and TM polarizations at different angles of the MA were also verified by measurements, via the rotation of the fabricated prototype at different angles for both the TE- and TM-polarized wave. The performance comparison showed good agreement between simulated and measured results. It is concluded that the desired absorption band can be attained by suitably tailoring the absorber’s constituents. Furthermore, these results validated that the proposed absorber would be useful for 5G communication applications, especially for the absorption of frequencies in 5G massive MIMO antenna arrays, to avoid the unwanted near-field interference.

## Figures and Tables

**Figure 1 sensors-22-03764-f001:**
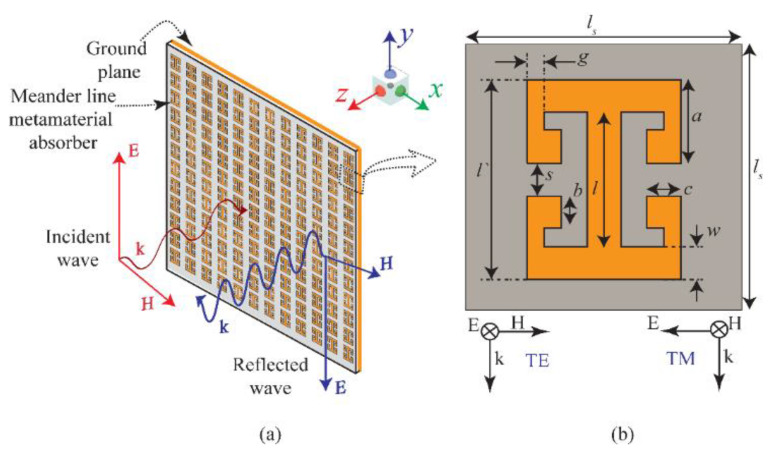
(**a**) Schematic representation of the finite absorber sheet consisting of the proposed meander-line-based 5G absorber. (**b**) Layout of the unit cell having optimized values as *l_s_* = 8 mm, *l′* = 6 mm, *l_w_* = 4.5 mm, *l* = 4 mm, *w* = 1 mm, *a* = 1.5 mm, *b* = 1 mm, *c* = 1 mm, *g* = 0.5 mm, and *s* = 1 mm.

**Figure 2 sensors-22-03764-f002:**
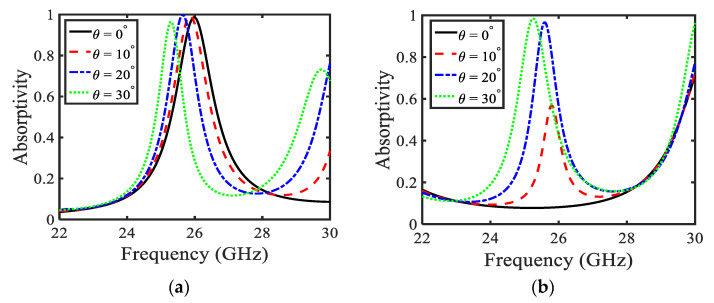
Absorption of the proposed MA keeping substrate thickness *t* = 0.8 mm (**a**) TE polarization and (**b**) TM polarization.

**Figure 3 sensors-22-03764-f003:**
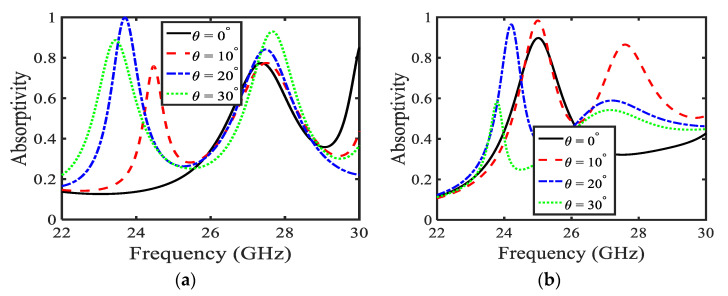
Absorption of the proposed MA keeping substrate thickness *t* = 1.2 mm for (**a**) TE polarization (**b**) TM polarization.

**Figure 4 sensors-22-03764-f004:**
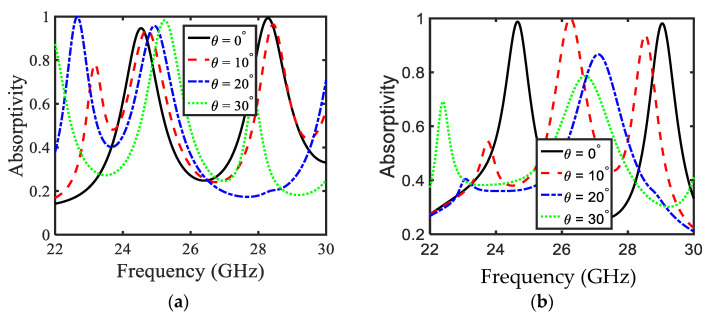
Absorption of the proposed PA keeping substrate thickness *t* = 1.6 mm for (**a**) TE polarization (**b**) TM polarization.

**Figure 5 sensors-22-03764-f005:**
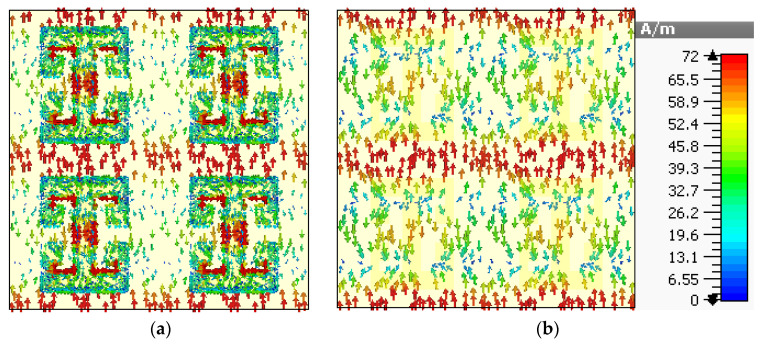
Surface current density for 24.7 GHz (**a**) top metasurface and (**b**) bottom surface.

**Figure 6 sensors-22-03764-f006:**
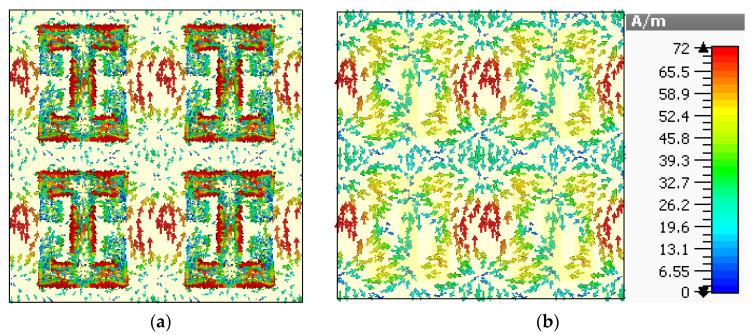
Surface current density for 28.3 GHz (**a**) top metasurface (**b**) bottom surface.

**Figure 7 sensors-22-03764-f007:**
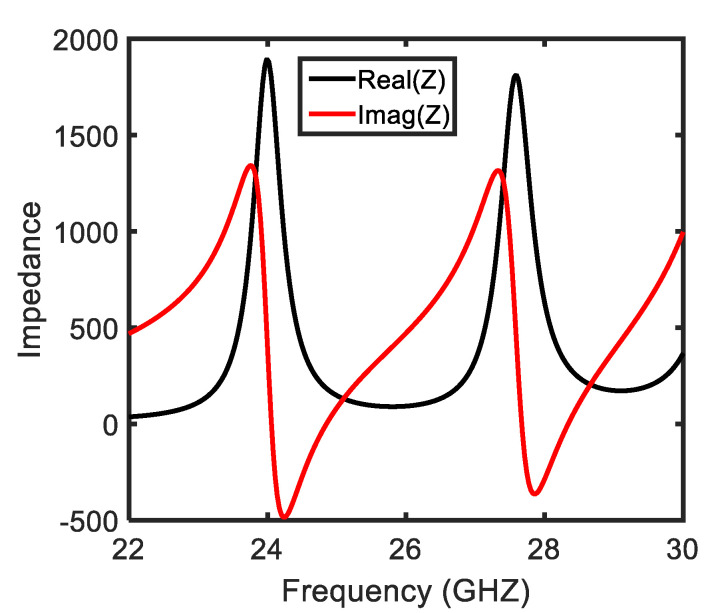
Effective impedance of the metamaterial absorber.

**Figure 8 sensors-22-03764-f008:**
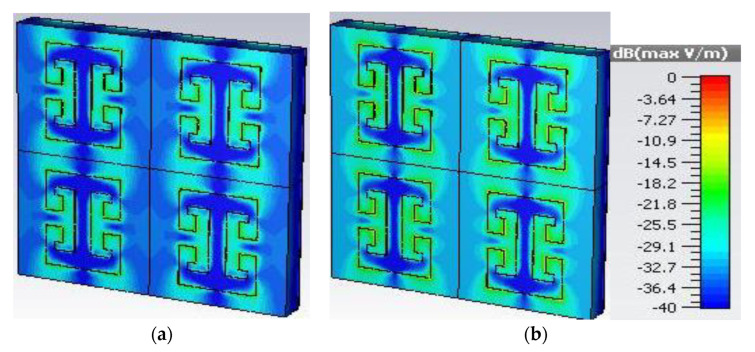
Simulated electric field distribution for 24.5 GHz for (**a**) TE polarized wave and (**b**) TM polarized wave.

**Figure 9 sensors-22-03764-f009:**
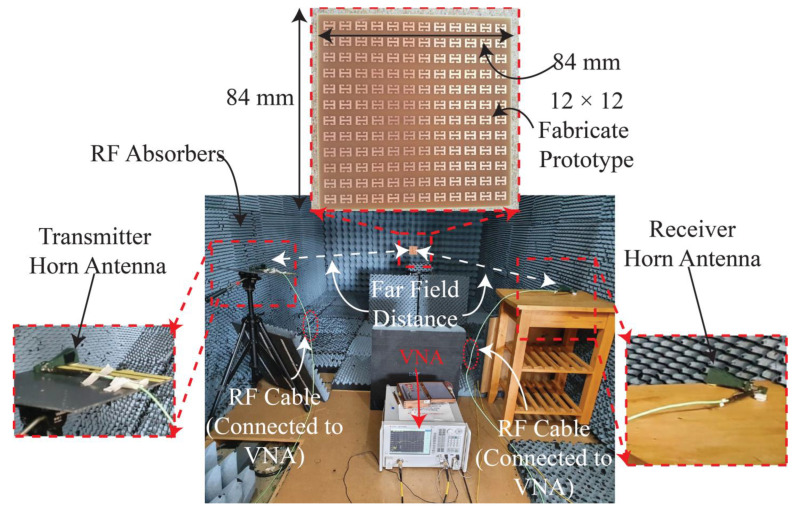
A photograph of the fabricated prototype and experimental setup used for the measurement of the proposed MA.

**Figure 10 sensors-22-03764-f010:**
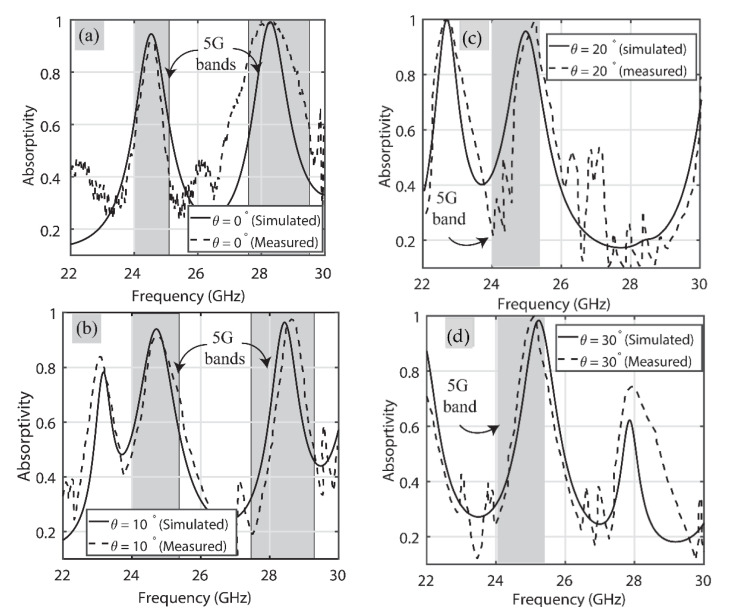
Performance comparison of the proposed MA for TE polarization at (**a**) 0°, (**b**) 10°, (**c**) 20° and (**d**) 30°.

**Figure 11 sensors-22-03764-f011:**
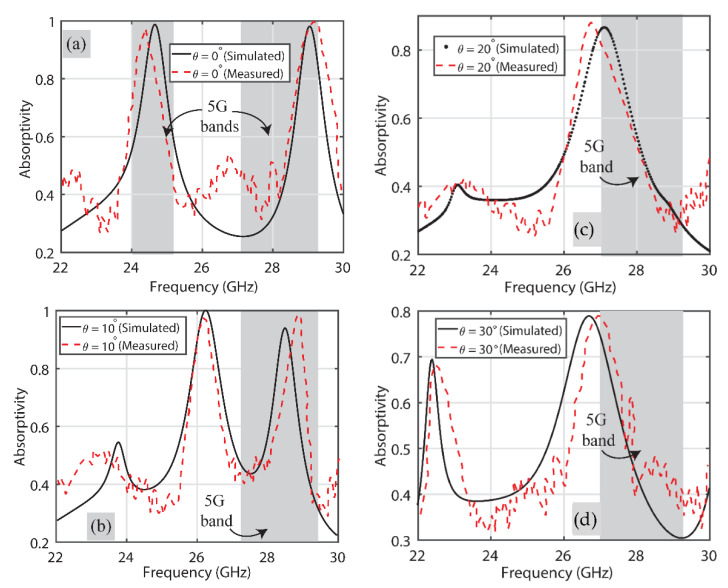
Performance comparison of the proposed MA for TM polarization at (**a**) 0°, (**b**) 10°, (**c**) 20° and (**d**) 30°.

## Data Availability

Not applicable.

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
