# Peer review of "A Novel Meander Line Metamaterial Absorber Operating at 24 GHz and 28 GHz for the 5G Applications"

_sensors, 2022, doi:10.3390/s22103764_

Round 1
Reviewer 1 Report
The authors have presented an interesting MA design for 5G frequencies at 24 GHz and 28 GHz. Overall the manuscript shows significant novelty for publication. The manuscript reads well and authors have provided sufficient analysis which makes it useful for general design of the absorber at other frequencies.
I only have couple of minor comments which authors should address.
- Authors have presented the absorptivity of the proposed MA at different incidence angle for TE and TM polarization in different figures across the manuscript and the absorptivity at 0° incidence angle is significantly shifted for the two polarizations. It is not clearly explained what causes this significant frequency shift for incidence angle less than 10°. Please justify this effect with more details.
- Although authors mention that it is the first time MA is applied for 5G frequencies still I believe it is good idea to present a table comparing other absorbers and show why the proposed absorber is novel than the other MA in the literature. Please include a table comparing the proposed MA with the state-of-the-art designs for absorbers.
Author Response
Authors Reply to Review Report (Reviewer 1)

Reviewer 2 Report
- In line 139, The design consists of symmetric pairs of meander lines (four meander lines) connected with an I-shape TL.......But the reviewer didn't found four meander lines. Here, four segments formed a meander line.
- In line 134, By putting the values of parameters shown in Figure 1 (b), the obtained number of turns are 3.30 and 3.35 at 24 GHz and 28 GHz, respectively. Consequently, the closest 135
larger integer of four meander lines is chosen in the absorber geometry to achieve unity absorption at the band of interest.....
But were are four turns of the meander lines.... It is confiusing to the reviewer. - The simulation setup should be included.
- How the author set TE and TM modes and match with Simulation setup? There should be a discussion.
- In Figure 9, the author should provide clear image showing S21 data in VNA.
- A comparison with existing works should be included.
Author Response
Authors Reply to Review Report (Reviewer 2)

Reviewer 3 Report
The authors present a very interesting work on a novel dual band metamaterial-based 5G absorber, and provide thorough analyses that is also proved by actual fabrication. The flow is smooth and results are presented clearly.
I have a minor comments regarding the actual device integration. Could the authors add more details on the fabrication? What is the technique applied for fabricating the actual metamaterial device? Please also add more details to explain the factors causing differences between simulated and measured absorptivity (Figure 11).
Author Response
Authors Reply to Review Report (Reviewer 3)

Round 2
Reviewer 2 Report
The author should take care on the quality of the figures.
Author Response
Thank you for your positive review and for your comment.
Figure 9, which is the main figure of the article, was enlarged for enhanced clarity.
